# FedGRec: Federated Graph Recommender System with Lazy Update of Latent Embeddings

## Abstract

Recommender systems are widely used in industry to improve user experience. Despite great success, they have recently been criticized for collecting private user data. Federated Learning (FL) is a new paradigm for learning on distributed data without direct data sharing. Therefore, Federated Recommender (FedRec) systems are proposed to mitigate privacy concerns to non-distributed recommender systems. However, FedRec systems have a performance gap to its non-distributed counterpart. The main reason is that local clients have an incomplete user-item interaction graph, thus FedRec systems cannot utilize indirect user-item interactions well. In this paper, we propose the Federated Graph Recommender System (FedGRec) to mitigate this gap. Our FedGRec system can effectively exploit the indirect user-item interactions. More precisely, in our system, users and the server explicitly store latent embeddings for users and items, where the latent embeddings summarize different orders of indirect user-item interactions and are used as a proxy of missing interaction graph during local training. We perform extensive empirical evaluations to verify the efficacy of using latent embeddings as a proxy of missing interaction graph; the experimental results show superior performance of our system compared to various baselines.

## 1 Introduction

Recommender systems play an essential role in reducing information overload in the current era of information explosion. A recommender system predicts a small set of candidates in which a user may be interested from a large number of items. Collaborative Filtering (CF) [11] is one of the most successful approaches to making recommendations. CF is based on the idea that users with a similar interaction history tend to share interests in items. Naturally, CF is highly dependent on collecting user behavior data. Gathering such information undermines the privacy of the user. To alleviate this challenge, researchers exploited the idea of Federated Learning (FL) and developed Federated Recommender Systems (FedRec). In a FedRec system, users keep its data locally and only share the model with the server. FedRec systems mitigate privacy concerns, but still have a performance gap with non-distributed recommender systems [1, 13]. In a FedRec system, a user performs local training with its own interaction data and cannot access the data of other users. With this incomplete interaction graph, the learned model generally cannot capture indirect user-item interactions well. In contrast, non-distributed recommender systems [18, 22] have access to the whole interaction graph and can capture such indirect interaction with various techniques such as graph embedding. Therefore, it is essential to develop a technique to mitigate the bias caused by the incomplete local interaction graph so that FedRec systems can better capture indirect interaction. In this paper, we take one step forward and propose the Federated Graph Recommender System (FedGRec), which can take advantage of indirect interaction efficiently.

Recently proposed federated recommender systems either abandon taking advantage of indirect interactions [1, 6, 45], or rely on complicated cryptography techniques [48] to access data from other users. In particular, [48] proposed FedGNN, which adapted graph neural network (GNN)-based recommender systems to the FL setting. In FedGNN, a user can request embeddings of its neighbors using encryption techniques. However, this is achieved at high cost. First, it assumes the existence of a trusted third party; second, this request needs a large amount of computing power for expensive Homomorphic Encryption [33] operations. Furthermore, even at this high cost, FedGNN only exploits first-order user-item interactions, *i.e.* the direct neighbors of a user, while in non-distributed GNN-based models, second-order interactions (users that interact with same items) and even higher-order indirect interactions are exploited. To alleviate the limitations of FedGNN and fully exploit indirect user-item interactions, we propose our FedGRec system, and the key feature of our system is to explicitly store latent embeddings of users and items. The concept of latent embedding of a user/item stems from the non-distributed GNN-based recommender system. Non-distributed GNN-based recommender systems [13, 46] usually have an embedding layer and multiple embedding propagation layers. The embedding layer encodes users and items to obtain a vector representation of them. Embedding propagation layers refine user/item embeddings sequentially. Each embedding propagation layer linearly combines neighbor embeddings of the last layer. Finally, embedding and output of embedding propagation layers are combined (such as average) as the final representation of a user/item. In fact, the output of the embedding propagation layers encodes different orders of user-item interactions. For ease of discussion, we denote them as latent embeddings. Our system is built on using latent embeddings as a proxy of the miss interaction graph during local training.

In our FedGRec system, users store their embeddings, and the server stores embeddings for all items as normal federated recommender systems. In addition, users also keep their latent embeddings, and the server has latent item embeddings. The training process includes two parts: the optimization of user/item embeddings and the optimization of latent user/item embeddings. First, assume that latent embeddings encode indirect user-item interactions; then users update user/item embeddings by treating latent embeddings as constants for multiple training steps locally. Next, the latent user/item embeddings are updated only in the server synchronization step based on the current user/item embeddings. Note that latent user/item embeddings are fixed during users' local training, which differs from that in the non-distributed setting. In the non-distributed setting, embedding propagation performed in real time, *i.e.* the latent embeddings encode up-to-date indirect user-item interaction information. In contrast, we use fixed latent user/item embeddings during local training due to communication constraints. This makes the information encoded in these latent embeddings stale. However, we empirically show that the stale latent embeddings are still useful in capturing indirect interaction. We verify their efficacy through extensive empirical studies. Finally, our system preserves the privacy of users. During the whole training process, only the item embeddings are transferred between users and the server, in addition, we take advantage of the secure aggregation technique [5]. Secure aggregation is a privacy-preserving technique that allows the server to get the sum of user updates without knowing individual details. Finally, we summarize the contributions of our paper as follows:

1. We propose a novel Federated Graph Recommender System (FedGRec) that effectively uses the indirect user-item interactions;

2. We design and store latent embeddings to encode the indirect user-item interaction. Latent embeddings are a proxy for absent neighbors during local training. We also propose a lazy way to update these latent embeddings;

3. Our new system is evaluated via extensive experimental studies, and results show the superior performance of our system compared to various baselines.

**Organization:** The remainder of this paper is organized as follows: In Section 2, we formally introduce our new Federated Graph Recommender System (FedGRec); In Section 3, we perform experiments to verify the effectiveness of our system; In Section 4, we conclude and summarize the paper. The discussion of some related works can be found in Appendix A.

## 2 FedGRec : A Novel Federated Graph Recommender System

In this section, we introduce our **Fed**erated **G**raph **Rec**omender System (**FedGRec**). We consider the horizontal federated recommender systems [53]: there is a server and $M$ users $\mathcal{U} = \{u_1, u_2, ..., u_M\}$.

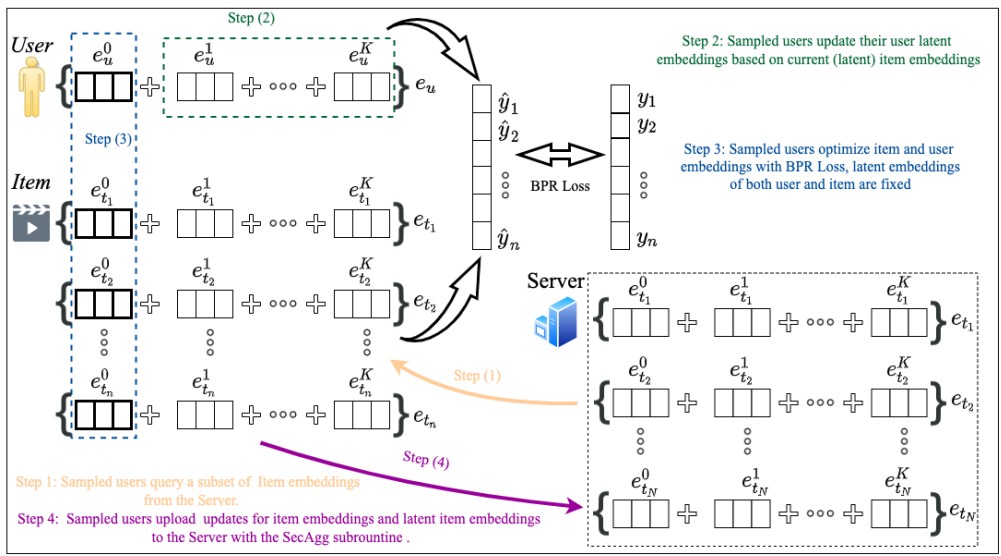

Figure 1: The framework of our FedGRec system. The server stores the item embeddings and latent item embeddings, while each user keeps its own user embedding and latent user embeddings. Each training epoch is composed of four main steps: (1) the sever randomly sample a subset of users, and each sampled user queries a subset of n items and their (latent) item embeddings from the server; (2) the user updates its latent user embedding based on the new (latent) item embeddings; (3) the user optimizes item and user embeddings with BPR loss; (4) the user uploads updates back to the server, and the server aggregates updates from all users.

Users interact with a common set of $N$ items $\mathcal{T} = \{t_1, t_2, ..., t_N\}$. More specifically, the user $u$ interacts with a subset of $|\mathcal{N}_u|$ items $\mathcal{N}_u = \{t_{u,1}, t_{u,2}, ..., t_{u,|\mathcal{N}_u|}\}$, and we have $\mathcal{T} = \bigcup_{u \in \mathcal{U}} \mathcal{N}_u$. (To protect user privacy, user interaction data do not leave the local device.) In non-distributed graph recommender systems, there is the input embedding layer and multiple embedding propagation layers. Embedding propagation layers are used to refine embeddings with high-order user-item interaction information. Resemble the design in the non-distributed setting, we let each user (item) be represented by a learnable embedding vector $e_u^0 \in \mathcal{R}^d$ ($e_t^0 \in \mathcal{R}^d$). Furthermore, the user (item) also keeps latent embeddings: $e_u^k \in \mathcal{R}^d$ ($e_t^k \in \mathcal{R}^d$) ($k \geq 1$). Latent embeddings are similar to the embedding propagation layers in the non-distributed setting and encode the indirect (high-order) user-item interaction information. In non-distributed recommender systems, latent embeddings can be computed in real-time based on the whole user-item graph. However, in FL, the interaction graph is incomplete for each client; instead we perform a lazy update to the latent embeddings.

More precisely, the FedGRec system training procedure is divided into two parts as shown in Figure 1 around the use of latent embeddings: local training with fixed latent embeddings and lazy update of (latent) user/item embeddings. In Figure 1, Step 3 corresponds to the local training, and Steps 2 and 4 correspond to the latent embedding update part. More specifically, at each epoch, a subset of clients is selected to perform training, and these clients query (a subset of) item embedding and item latent embeddings from the server (Step 1). Then, all selected users update their user latent embeddings following the message general passing procedure in the graph neural network (Step 2). Next, the user (item) embeddings are optimized under some objective, *e.g.* the BPR loss [37], where the user (item) latent embeddings are used as the proxy of the embedding propagation layers of the non-distributed setting (Step 3). Note that user (item) latent embeddings are fixed during Step 3 as we cannot get a real-time update from other clients for both privacy and communication issues. Although the latent embeddings are stale, they include useful indirect connection information from other clients, and we provide empirical evidence of its efficacy. Finally, in Step 4, clients upload the new item embeddings and item latent embeddings to the server, and we protect user privacy with the secure-aggregation technique (more details in Appendix B). On the server side, the server aggregates updates from all sampled clients and updates the item embeddings.

**Algorithm 1** **F**ederated **G**raph **Rec**ommender System (**FedGRec**)

---

1: **Input:** Learning Rate $\alpha$ and $\beta$; Number of global epochs $T$; Number of Latent Embeddings $K$; Number of local iterations $\tau$; Number of sampled Users per epoch $S$; Noise scale $\sigma$;
2: Warm-up Phase:
3: Server queries item connection information $|\mathcal{N}_t|$ with Eq. (9);
4: Server initializes $e_t^0$ (for $t \in \mathcal{T}$) with Gaussian Noise $\mathcal{N}(0, \sigma)$;
5: Every client $u$ for $u \in \mathcal{U}$ initializes its user embedding $e_u^0$ with Gaussian Noise $\mathcal{N}(0, \sigma)$;
6: **for** k in 1 **to** K **do**
7:     Each client requests $e_t^{k-1}$ for $t \in \mathcal{T}$ and computes $e_u^k$ based on Eq. (6);
8:     Each client uploads $\widetilde{E}_u^{k-1}$ to the server and the server computes $E_{\mathcal{T}}^k$ with Eq. (8);
9: **end for**
10: Training Phase:
11: **for** $l = 0$ **to** $T - 1$ **do**
12:     The server samples a random subset of clients $\widetilde{U}$ and $|\widetilde{U}| = S$;
13:     **for** u in $\widetilde{U}$ in parallel **do**
14:         Query the (latent) item embedding $e_t$ of a randomly selected item set $\widetilde{\mathcal{T}}_u$;
15:         Update user latent embedding with $e_t$ based on Eq. (6), denote the updated latent user embedding as $\tilde{e}_u^k, k \in [1, \dots, K]$;
16:         Set $(e_u^0)^0 = e_u^0$ and $(e_t^0)^0 = e_t^0$;
17:         **for** i in 1 **to** $\tau$ **do**
18:             $(e_u^0)^{i+1} = (e_u^0)^i - \beta \nabla_{e_u^0} L_{BPR,u}((e_u^0)^i, (e_t^0)^i; \mathcal{B})$
19:             $(e_t^0)^{i+1} = (e_t^0)^i - \beta \nabla_{e_t^0} L_{BPR,u}((e_u^0)^i, (e_t^0)^i; \mathcal{B})$
20:         **end for**
21:         Set $\tilde{e}_u^0 = (e_u^0)^\tau$ and $\tilde{e}_t^0 = (e_t^0)^\tau$;
22:         Upload $\tilde{e}_t^0 - e_t^0$ and $\tilde{e}_u^k - e_u^k, k \in [0, \dots, K-1]$ to the server with the *SecAgg* subroutine, the server updates the latent item embeddings with Eq. (10) and (11).
23:     **end for**
24: **end for**

---

Note that our FedGRec system is agnostic to specific message-passing mechanisms. In Appendix C, we introduce an instantiation of our FedGRec based on the LightGCN [46] system, which has been shown to be successful in training graph-based implicit recommendation in the non-distributed setting. In Algorithm 1, we provide a pseudocode of our FedGRec system. Lines 3-9 are the warm-up phase, which calculates the item connection information $|\mathcal{N}_t|$ and initializes the user and item (latent) embeddings; then lines 11-24 are the training phase (Figure 1). During local training (lines 18-19), we use the SGD update rule and the BPR loss [37] as an example. For privacy protection and communication costs, see Appendix C.3 for more details.

## 3 Experiments

In this section, we empirically validate the efficacy of our FedGRec system through extensive experiments. We simulate the Federated Learning environment based on the *Distributed* Library of Pytorch [34], and experiments are conducted on 4 servers with 4 NVIDIA P40 GPUs each.

### 3.1 Experimental Settings

**Datasets.** We choose three widely used benchmark datasets in non-distributed recommendation: Gowalla [25], Yelp2018 [46] and Amazon-Book [12]. The statistics of these datasets are shown in Table 2 of the Appendix D. We use $Recall@20$ and $NDCG@20$ as the metric (a detailed description of the two metrics is provided in Appendix D). We follow the train/test split provided by [13].

**Baselines.** We compare our FedGRec system with the baselines of the non-distributed and federated recommender system baselines. For non-distributed recommender systems, we compare with the following state-of-the-art recommender systems: Mult-VAE [26], NGCF [46] and LightGCN [13]. Mult-VAE is a collaborative filtering method based on variational autoencoder (VAE) that gets competitive results over many datasets. NGCF and LightGCN are two graph-based recommender systems and are closely related to our FedGRec system. Next, for the federated baselines, we

Table 1: Performance Comparison of FedGRec and Baselines (Recall and NDCG)

| | Dataset | Gowalla | | Yelp2018 | | Amazon-Book | |
|---|---|---|---|---|---|---|---|
| | Method | Recall | NDCG | Recall | NDCG | Recall | NDCG |
| **Non-distributed RecSys** | Mult-VAE | 0.1641 | 0.1335 | 0.0584 | 0.0450 | 0.0407 | 0.0315 |
| | NGCF-1 | 0.1556 | 0.1315 | 0.0543 | 0.0442 | 0.0313 | 0.0241 |
| | NGCF-2 | 0.1547 | 0.1307 | 0.0566 | 0.0465 | 0.0330 | 0.0254 |
| | NGCF-3 | 0.1570 | 0.1327 | 0.0566 | 0.0461 | 0.0344 | 0.0263 |
| | LightGCN-1 | 0.1755 | 0.1492 | 0.0631 | 0.0515 | 0.0384 | 0.0298 |
| | LightGCN-2 | 0.1777 | 0.1524 | 0.0622 | 0.0504 | **0.0411** | 0.0315 |
| | LightGCN-3 | **0.1823** | **0.1555** | **0.0639** | **0.0525** | 0.0410 | **0.0318** |
| **Federated RecSys** | FCF | 0.0703 | 0.0588 | 0.0282 | 0.0235 | 0.0112 | 0.0088 |
| | FedMF | 0.0727 | 0.0583 | 0.0250 | 0.0207 | 0.0100 | 0.0079 |
| | FedeRank | 0.1440 | 0.1164 | 0.0503 | 0.0405 | 0.0287 | 0.2204 |
| | FedNCF | 0.0754 | 0.0575 | 0.0271 | 0.0218 | 0.0093 | 0.0075 |
| | FedGNN | 0.1556 | 0.1211 | 0.0543 | 0.0396 | 0.0229 | 0.0208 |
| | FedGNN + BPR | 0.1676 | 0.1362 | 0.0601 | 0.0498 | 0.0339 | 0.0269 |
| | FedGRec-1 | **0.1712** | 0.1376 | 0.0598 | 0.0491 | 0.0342 | 0.0268 |
| | FedGRec-2 | 0.1695 | **0.1412** | 0.0607 | 0.0497 | **0.0361** | **0.0285** |
| | FedGRec-3 | 0.1654 | 0.1362 | **0.0615** | **0.0503** | 0.0333 | 0.0262 |

compare with the recently proposed methods: FCF [32], FedMF [6], FedNCF [35], FedGNN [48] and FedeRank [2]. FCF, FedMF, and FedeRank are matrix factorization-based methods where FCF/FedMF uses the MSE loss, while FedeRank uses the BPR loss. FedNCF adapts the NCF [15] to the FL setting, FedGNN is a recently proposed graph-based recommender system.

**Parameter settings.** In all our experiments, the embedding size $d$ is fixed at 64 for all methods and the user/item embeddings are initialized with the normal distribution (as in the Pytorch implementation of [13]). By default, we run the $T = 10^5$ epochs. During each training epoch, we randomly select 400 users by default. For each user, it queries all its positive items and a random subset of negative items of size 2048. In local training, we use Adam optimizer with a learning rate of 0.001. For other hyperparameters, we perform a grid search for each method and report the best results. For Mult-VAE, NGCF, and LightGCN, we use the hyperparameter settings in [13]. For baselines of the federated recommender systems: In FCF and FedMF, we choose the confidence parameter $\alpha = 1$, $L_2$ regularization parameter $\lambda = 10^{-3}$, local iterations $\tau = 1$; in FedNCF, we implement the Fed-NeuMF variant. We use a three-layer MLP with hidden units [32, 16, 8], $L_2$ regularization parameter $\lambda = 10^{-4}$. In FedGNN and FedeRank, we follow the parameter setting in the original paper. For our FedGRec method, we choose $L_2$ regularization parameter $\lambda = 10^{-4}$ and local iterations $\tau = 10$. For the latent embedding combination coefficient $\alpha_k$, we choose $1/(K + 1)$.

Finally, for graph-based methods (NGCF and LightGCN), we vary the number of embedding propagation layers and use method-$k$ to represent $k$ layers. The FedGNN method only supports one-layer graph neural network, so we omit the post-fix for it. For our method, we vary the number of latent embeddings and use FedGRec-$k$ to represent using $k$ latent embedding vectors.

### 3.2 Performance Evaluations

The full experimental results are shown in Table 1. Compared to non-distributed recommender systems, our FedGRec outperforms Mult-VAE and NGCF and is comparable to the LightGCN method. This shows that it is reasonable to use latent embeddings as an alternative to the exact neighbor-user/item embeddings.

Next, we compare our FedGRec with the baselines of the federated recommender system. First, for the three matrix factorization-based baselines: FCF, FedMF, and FedeRank, our FedGRec outperforms them by a great margin. In particular, the FedeRank method can be viewed as a special case of our FedGRec system where indirect interaction is not used, and the superior performance of our system validates the efficacy of using high-order indirect interaction in federated recommender systems. Furthermore, we plot the NDCG/Recall curve for FedeRank and our FedGRec in Figure 2. We observe that FedeRank converges fast in the early training stage (around the first 5000 epochs), but

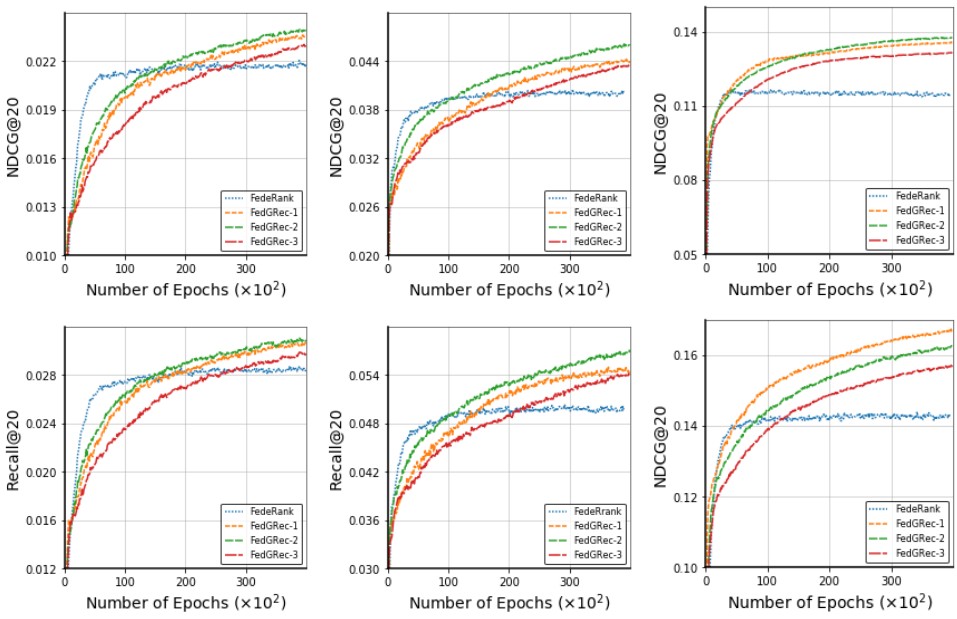

Figure 2: NDCG@20 and Recall@20 over different number of latent embeddings. Results correspond to the yelp2018 dataset, the Amazon-Book dataset and the Gowalla dataset from top to bottom.

it then overfits to the $\text{zero}_{th}$ order user-item connection and converges to a sub-optimal point. This phenomenon further demonstrates the efficacy of using latent embeddings in our FedGRec system. Next, for FedNCF, it only gets performance comparable to FCF/FedMF. The main reason for this underperformance is the heterogeneity of user-interaction distributions. Due to the heterogeneity, the neural networks severely overfit to the local distribution.

Finally, for experiments related to FedGNN, we report results for FedGNN as in the original paper, and also a variant where the MSE loss objective is replaced with the BPR loss objective. We denote this variant by FedGNN+BPR. We can see a performance boost of FedGNN+BPR compared to the original FedGNN. In fact, FedGNN + BPR gets an equivalent performance as our FedGRec-1 variant, which is reasonable since FedGNN exploits the first-order user-item interaction. However, our FedGRec is still advantageous over it. First, the best results are obtained in FedGRec-2 / FedGRec-3 in most cases, *e.g.* FedGRec-3 has the best performance in the Yelp2018 dataset. In contrast, the FedGNN method can only exploit the first-order interaction. Second, note that FedGNN uses a user-item graph expansion operation to get neighbors of a user anonymously, while the expansion operation requires time-consuming cryptography techniques to protect user privacy. However, our FedGRec does not need this operation, and we only use latent embedding information in training. So our FedGRec is much more efficient. For some ablation study and hyper-parameter analysis, please see Appendix D.1.

## 4 Conclusion

In this paper, we propose a novel federated graph recommender system (FedGRec). Our system effectively exploits indirect user-item interaction to improve recommendation performance. We explicitly store the latent user and item embeddings that encode the indirect user-item interaction information. We propose using a lazy update to these latent embeddings and using the secure aggregation technique to protect user privacy. Experiments conducted over common recommendation benchmarks show that our system achieves competitive performance with non-distributed Graph Neural Network based recommender systems and superior performance over other federated recommender systems.

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

# A   Related Work

The study of Recommender Systems dates back to the 1990s [11]. Most recommender systems aim to develop representations for users and items. A classic approach is based on matrix factorization: we associate the embedding vectors with the one hot user(item) ID [37, 18, 22], and then take inner products between the embeddings of the user and the item to match the ratings. Later, researchers pooled interacted item embeddings to augment user representation, such as FISM [18], SVD++ [22]. Furthermore, the graph neural network is also used to generate representations [4, 46, 13, 7]. Unlike learning representations, another line of work focuses on modeling interactions. There are two limitations of the inner product approach [15]. First, it does not satisfy the triangle inequality, so the propagation of similarity is not good; second, the linear nature of the inner product constrains its ability to model complicated interactions. Therefore, the researchers propose to use other metrics such as $L_2$ distance [16], and deep neural networks [14]. Although most of the recommender systems only use user-item interaction data, some additional data can boost model performance if available. These models can be divided into two categories: Content-based models [36, 42, 8, 50, 44] and context-based models [41, 28]. Content-based models use additional user features (items). In contrast, context-based models use auxiliary information from the interaction. [49] and [51] provide a detailed review of the recommender systems.

Federated learning [29] is a promising distributed data mining paradigm in which a server coordinates a set of clients to learn a model. A widely used algorithm for FL is the FedAvg [29] algorithm, where clients receive the up-to-date model from the server at the start of each epoch and then train the model locally for several iterations and upload the new model back to the server. There are three main challenges in FL: data heterogeneity, high communication cost, and user privacy. Some variants of FedAvg are proposed to address heterogeneity [19, 24, 39, 54, 31, 23]. To reduce the cost of communication, various compression techniques are applied, such as quantization [47, 27] and sparsification [40, 20, 38, 17]. Regarding user privacy, although the server cannot see the data directly, it is possible to recover the data based on model updates with a model inversion attack [10]. Therefore, some cryptography techniques are applied, such as homomorphic encryption [33], differential privacy [29] and multiparty secure computation [43] *etc.*. A simple but effective technique to defend a malicious server is the secure aggregation [5, 3, 52] technique. With this technique, the server aggregates updates from clients without knowing the input of each client.

More recently, Recommender systems have been considered in the federated learning setting (FedRec) [1, 6, 30, 52, 35, 45, 21, 32, 9, 2]. In particular, [1] applied the matrix factorization approach to FL. It used the Alternating Least Squares (ALS) algorithm. At each epoch, each client computes the optimal user embedding, then calculates the item embedding gradients, and uploads them to the server. Finally, the server aggregates the gradients from all clients to update the embeddings of the items. The above approach directly transfers the gradients of the item embeddings, which has the risk of leaking private ratings; Some privacy preservation techniques [45, 30, 52] are exploited to mitigate this risk [6]. In addition to classic matrix factorization-based approaches, deep neural collaborative filtering techniques are also adapted to the FL setting [35]. In [35], the authors proposed a two-stage training framework. In the first stage, item embeddings are learned with self-supervised learning. Then, in the second stage, a federated neural recommender system is learned with the help of differential privacy. Graph-based recommender systems have gained state-of-the-art performance in the non-distributed setting. However, it is not trivial to adapt them to the FL setting. In FL, each client only has a subgraph. Recent work [48] proposed to obtain the embeddings of neighboring users using the homomorphic encryption technique. Our paper also considers graph-based FedRec. However, we do not require the time-consuming homomorphic encryption technique, but we use the fact that only aggregated representations are needed in the training. The survey paper [53] provides a good overview of the problem of federated recommender systems.

# B   Preliminaries

**Graph Recommender Systems.** By exploiting indirect user-item interactions, graph-based recommender systems have gained state-of-the-art recommendation performance in the non-distributed setting. LightGCN [46] is a recently proposed graph recommendation system. It simplifies the classical graph neural network by removing the transformation matrix and the nonlinear activation function. The system includes two types of layer: the input embedding layer and the embedding propagation

layer. More precisely, there is one embedding layer which initializes (item) user embeddings, and several embedding propagation layers which refine embeddings with high-order user-item connectivity relations. Suppose that there are $K$ embedding propagation layers; then the $k_{th}$ ($k \in [0 \ldots, K-1]$) embedding propagation layer performs the following rule:

$$e_u^{k+1} = \sum_{t \in \mathcal{N}_u} \frac{e_t^k}{\sqrt{|\mathcal{N}_t|}\sqrt{|\mathcal{N}_u|}}; \; e_t^{k+1} = \sum_{u \in \mathcal{N}_t} \frac{e_u^k}{\sqrt{|\mathcal{N}_t|}\sqrt{|\mathcal{N}_u|}} \tag{1}$$

$\mathcal{N}_u$ is the set of connected items of the user $u$ and $\mathcal{N}_t$ is the set of connected users of the item $t$. The final representation (embedding) of a user/item is a weighted average of the output of these embedding layers, *i.e.*:

$$e_u = \sum_{k=0}^{K} \alpha_k e_u^k; \; e_t = \sum_{k=0}^{K} \alpha_k e_t^k \tag{2}$$

where $\alpha_k$ are weights. Then we calculate the inner product between the user and the item representation as a measure of their affinity: $\hat{y}_{ut} = \langle e_u, e_t \rangle$. During training, we optimize the user/item embedding so that $\hat{y}_{ut}$ is close to the true affinity. Various loss objectives could be used, such as the mean square error (MSE) loss and the Bayesian personalized ranking (BPR) loss [37]:

$$L_{MSE} = \sum_u \sum_{t \in \mathcal{N}_u} (1 - \hat{y}_{ut})^2 \tag{3}$$

and

$$L_{BPR} = \sum_u \sum_{i \in \mathcal{N}_u} \sum_{j \notin \mathcal{N}_u} S^+(\hat{y}_{uj} - \hat{y}_{ui}) \tag{4}$$

where $S^+(x) = \log(1 + e^x)$ is the Softplus function. In the MSE loss (3), we select user/item pairs $(u, t)$ with interaction and denote their affinity score as one, then we minimize the $L_2$ error of the predicted affinity $\hat{y}_{ut}$. Next, in the BPR loss (4), we learn embeddings such that the interacted user-item pairs remain close while the uninteracted pairs are far apart.

**Secure Aggregation.** Secure Aggregation [5] is a privacy-preserving aggregation technique widely used in FL. The technique can securely compute the sum of vectors without revealing the value of each vector. In Secure Aggregation, we add a mask to each vector: The mask hides the original information, but can be canceled when all vectors are added together. More formally, suppose that we have a set of users $u \in \mathcal{U}$ and that each user has a vector $x_u$. To calculate $\sum_{u \in \mathcal{U}} x_u$, we first generate a random seed $r_{u,v}$ for each pair of users $(u, v)$. Then the user $u$ reveals:

$$\tilde{x}_u = x_u + \sum_{v \in \mathcal{U}, v < u} PRG(r_{u,v}) - \sum_{v \in \mathcal{U}, v > u} PRG(r_{u,v})$$

Note we assume a total order of users for convenience. $PRG$ is short for Pseudo Random Generator. It is straightforward to see that $\sum_{u \in \mathcal{U}} \tilde{x}_u = \sum_{u \in \mathcal{U}} x_u$. As a result, the server recovers sum of vectors without knowing the value of $x_u$. In practice, we should consider the possibility of user drop-out, we then need additional random masks under this case. Various mechanisms are proposed [5, 3, 52], and we will not consider user dropout in our experiments for simplicity. The overall communication complexity of secure aggregation is at the same order of sending data in the clear. Note that the Secure Aggregation is relatively independent to our system design, so we will use it as a oracle subroutine in the remainder of the text and use $SecAgg(\cdot)$ to denote it.

## C  More Details of the FedGRec

This section introduces an instantiation of our FedGRec system based on the popular LighGCN [46] network. Appendix B introduce some background of LightGCN. In Appendix C.1, we show the local training procedures, *i.e.* Step 3 in Figure 1, next in Appendix C.2, we show the lazy update of latent embeddings, *i.e.* Steps 2 and 4 in Figure 1. Finally, Appendix C.3 analyze the privacy protection and communication cost of our system.

## C.1 Local Training with Fixed Latent Embeddings

In this subsection, we introduce local training procedures with fixed latent embeddings. As shown in Figure 1, the server has the (latent) item embeddings $e_t = \{e_t^k, k \in [0, \ldots, K]\}$ for $t \in \mathcal{T}$ and each user has its own (latent) embedding $e_u = \{e_u^k, k \in [0, \ldots, K]\}$ for $u \in \mathcal{U}$. Note that $K$ denotes the number of latent embeddings per user (item), and $K$ latent embeddings can encode user-item interactions up to order $K$. This is analogous to adopting a $K$-layer graph neural network in a non-distributed recommender system.

During each training epoch, the server randomly samples a batch $\widetilde{\mathcal{U}} \subset \mathcal{U}$ of $S$ users. As shown in Step 1 of Figure 1, each sampled user $u \in \widetilde{\mathcal{U}}$ randomly samples a subset $\widetilde{\mathcal{T}}_u \subset \mathcal{T}$ of items and requests their (latent) embeddings $\{e_t\}, t \in \widetilde{\mathcal{T}}_u$ from the server. Note that $\widetilde{\mathcal{T}}_u \neq N_u$, and the user samples both positive and negative items. This prevents the server from knowing the user's interaction history and damaging user privacy. After receiving (latent) item embeddings, the user optimizes the user and item embeddings with its local data. More precisely, the user $u$ optimizes the BPR loss:

$$L_{BPR,u}(e_u^0, e_t^0; \mathcal{B}) = \sum_{(j,i) \in \mathcal{B}} S^+ (\hat{y}_{ui} - \hat{y}_{uj}) \tag{5}$$

In practice, we add $L_2$ regularization to the above objective to avoid overfitting; we omit it here for simplicity. Furthermore, $\mathcal{B}$ is a mini-batch of positive and negative sample pairs $(j, i)$. $\hat{y}_{ui}$ and $\hat{y}_{uj}$ are estimated probabilities in which the user interacts with the items $t_j$ and $t_i$. Note that only $e_u^0$ and $e_t^0$, for $t \in \widetilde{\mathcal{T}}_u$, are learnable and latent embeddings are viewed as constants. To make it clearer, we can also rewrite the loss $L_{BPR,u}$ as a function of the user embedding $e_u^0$ and the item embedding $e_t^0$, $t \in \widetilde{\mathcal{T}}_u$ as follows:

$$L_{BPR,u} = \sum_{(j,i) \in \mathcal{B}} S^+ \left( \alpha_0^2 \left( e_u^0 \right)^T \left( e_{t_i}^0 - e_{t_j}^0 \right) + A e_u^0 + B \left( e_{t_i}^0 - e_{t_j}^0 \right) + C \right)$$

It is straightforward to derive the above formulation from Eq. (5), and we omit it because of space limitations. $A = \sum_{k=1}^{K} \alpha_k (e_{t_i}^k - e_{t_j}^k)$, $B = \sum_{k=1}^{K} \alpha_k e_u^k$, and $C = A \times B$. The user can optimize Eq. (5) with any optimizer such as the Adam optimizer. In practice, we optimize the objective Eq. (5) multiple steps before the user sends the updates back to the server. This is a common practice in FL to reduce communication costs and is also the main reason why real-time latent embeddings are not available.

## C.2 Lazy Update of (Latent) User/Item Embeddings

In the previous subsection, we assume access to the latent embeddings and ignore the update procedure of the latent embeddings. In this subsection, we discuss how we update latent embeddings so that they can encode indirect user-item interactions. The update of latent embeddings consists of two phases: the warm-up phase and the training phase. The warm-up phase is used to perform the initialization. The server initializes item embeddings $e_t = \{e_t^k, k \in [0, \ldots, K]\}$ for $t \in \mathcal{T}$, and each user $u$ initializes its embedding $e_u = \{e_u^k, k \in [0, \ldots, K]\}$ for $u \in \mathcal{U}$. Note that $\{e_u^0\}$ and $\{e_t^0\}$ are initialized directly *e.g.* with Gaussian noise, while the latent embeddings $\{e_u^k\}$ and $\{e_t^k\}$ for $k \geq 1$ are placeholders (initialized with 0). The exact values of the latent embeddings are jointly evaluated by the server and the users. More precisely, we perform $K$ successive rounds to evaluate latent embeddings. In the round $k$ ($k \in [1, \ldots, K]$), we evaluate the $k_{th}$ latent embedding based on the $(k-1)_{th}$ latent embedding. For the user $u$, it requests $(k-1)_{th}$ latent item embeddings (requests item embedding if $k = 0$) from the server and evaluates $e_u^k$ as follows:

$$e_u^k = \sum_{t \in \mathcal{N}_u} \frac{1}{\sqrt{|\mathcal{N}_t|}\sqrt{|\mathcal{N}_u|}} e_t^{k-1} \tag{6}$$

Although Eq. (6) only needs $t \in \mathcal{N}_u$, the user requests the whole set of item embeddings to avoid revealing to the server its interaction history. For the server, it evaluates $e_t^k$ for $t \in \mathcal{T}$. We use the matrix form here for clarity. First, each user $u$ generates an update matrix $\widetilde{E}_u^{k-1}$ as follows:

$$\widetilde{E}_u^{k-1} = Y_u^T \times D \left( \frac{1}{\sqrt{|\mathcal{N}_t|}\sqrt{|\mathcal{N}_u|}} \right) \times \left( e_u^{k-1} \right)^T \tag{7}$$

Recall that $Y_u$ is the row of the adjacency matrix $Y$ that corresponds to the user $u$. $D \in \mathbb{R}^{N \times N}$ is a diagonal matrix with the $t_{th}$ diagonal element as $\frac{1}{\sqrt{|\mathcal{N}_t|}\sqrt{|\mathcal{N}_u|}}$. In summary, user $u$ proposes updates for all connected items. Then the server aggregates $\widetilde{E}_u^{k-1}$ from all users with the secure aggregation subroutine:

$$E_{\mathcal{T}}^k = SecAgg\left(\widetilde{E}_u^{k-1}, u \in \mathcal{U}\right) \tag{8}$$

Note that $|\mathcal{N}_t|$ and $|\mathcal{N}_u|$ are normalizing factors that prevent the explosion of the embedding scale. Each user can calculate $|\mathcal{N}_u|$ directly with its own information. For $|\mathcal{N}_t|$, we obtain it in a way that preserves privacy with the subroutine $SecAgg$:

$$|N_t| = SecAgg\left(Y_u, u \in \mathcal{U}\right)_t \tag{9}$$

After $K$ rounds of running Eq. (6) and Eq. (8), we finish the warm-up phase and it is straightforward to verify that the latent embeddings $\{e_u^k, u \in \mathcal{U}\}$ and $\{e_t^k, t \in \mathcal{T}\}$ satisfy the Eq. (1).

In the training phase, user and item embeddings are updated during each epoch, as we discussed in Section C.1, latent embeddings should also be updated accordingly. During every training epoch, the user $u$ receives the (latent) item embeddings $\{e_t\}, t \in \widetilde{\mathcal{T}}_u$ from the server. The user first needs to update its latent user embeddings $e_u^k, k \in [1, \ldots, K]$ with the new (latent) item embeddings (step 2 in Figure 1). The update equation is the same as Eq. (6), furthermore, we can update all $K$ orders of latent embeddings within one round. We denote updated latent user embeddings as $\tilde{e}_u^k, k \in [1, \ldots, K]$. The user then optimizes both the user and the item embeddings following the steps of Section C.1 (step 3 in Figure 1). We denote updated user and item embeddings as $\tilde{e}_u^0$ and $\tilde{e}_t^0$, respectively. The last step is to send updates of (latent) item embeddings to the server (step 4 in Figure 1). For item embeddings, the user sends $\tilde{e}_t^0 - e_t^0$ and the server aggregates with the $SecAgg$ subroutine and then update the item embeddings $e_t^0$ as:

$$e_t^0 = e_t^0 + \alpha \times SecAgg(\tilde{e}_t^0 - e_t^0, u \in \widetilde{\mathcal{U}}) \tag{10}$$

while for latent item embeddings, the server updates the latent item embeddings $E_{\mathcal{T}}^k, k \in [1, \ldots, K]$ as follows:

$$E_{\mathcal{T}}^{k+1} = E_{\mathcal{T}}^{k+1} + \alpha \times SecAgg\left(Y_u^T \times D\left(\frac{1}{\sqrt{|\mathcal{N}_t|}\sqrt{|\mathcal{N}_u|}}\right) \times \left(\tilde{e}_u^k - e_u^k\right)^T, u \in \widetilde{\mathcal{U}}\right) \tag{11}$$

where $\alpha$ is the learning rate. In summary, latent user embeddings are updated when a user receives the new (latent) item embeddings. For latent item embeddings, a user proposes embedding updates to all its connected items if it is selected in a training epoch. We term this as a lazy update of latent embeddings. This is reflected in two ways: First, the latent embeddings are fixed when the user optimizes the objective Eq. (5) locally; Secondly, only active users update the latent embeddings during each training epoch.

Table 2: Statistics of the datasets

| Dataset | #Users | #Items | #Interactions | Density |
| --- | --- | --- | --- | --- |
| Gowalla | 29,858 | 40,981 | 1,027,370 | 0.00084 |
| Yelp2018 | 31,831 | 40,841 | 1,666,869 | 0.00128 |
| Amazon-Book | 52,643 | 91,599 | 2,984,108 | 0.00062 |

## C.3 Analysis of Privacy Protection and Communication Cost

User privacy protection is an important consideration in the design of the FL system. In our system, we protect the privacy of the user basically with the secure aggregation technique. During the whole training phase, the server only knows the aggregated information *e.g.*, the server knows $|\mathcal{N}_t|$ (the number of connected users per item), but it does not know the connection information of individual users. In addition, users request positive and negative items during training. This is required by the BPR loss, but it also hides user-connection information from the server.

Regarding communication cost, our FedGRec system requires the same order of communication as the simple Matrix Factorization approach [1]. For simplicity of discussion, suppose that all items

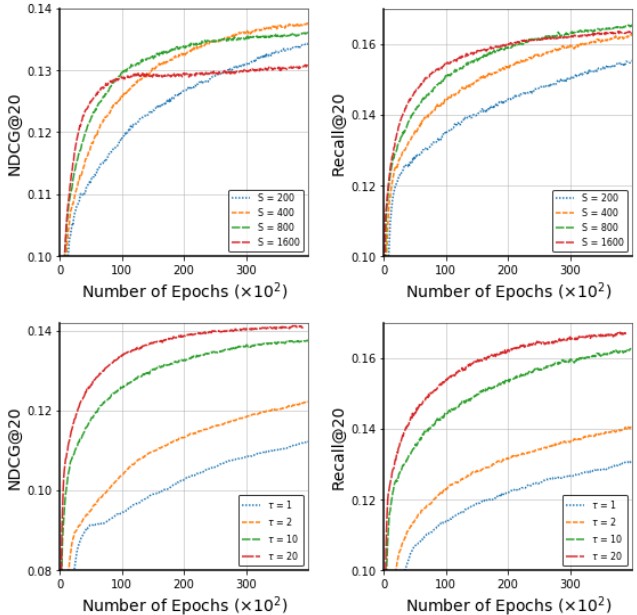

Figure 3: NDCG@20 and Recall@20 when we vary sampled users $S$ per training epoch (top row) and the number of local iterations $\tau$ (bottom row). Results are run over the Gowalla Dataset and we use two latent embeddings in training.

and users participate in the training every epoch and we ignore the extra communication cost caused by secure aggregation (communication complexity with secure aggregation is at the same order of sending data in the clear). First, the communication cost of a matrix factorization method [6, 32] is $O(TNMd)$ where $T$ is the total number of training epochs, $(M)$ $N$ is the number of (users) items and $d$ is the embedding dimension. For our system, in the initialization phase, we need to transfer the information on the order of $O(2KNMd)$, where $K$ is the number of latent embeddings. Then in the training phase, we need to transfer on the order of $O(2TKNMd)$. Therefore, the total communication cost is $O(KTNMd)$. Since $K$ is usually a small value (less than 5), our system achieves the same order of communication complexity as the matrix factorization method.

## D    More Details of Experimental Settings

The statistics of these datasets are shown in Table 2. We use metrics $Recall$ and $NDCG$ to evaluate our FedGRec system. Suppose that for each user $u$, and the set of its unconnected items is $\mathcal{T}_{test,u}$ (items not in the training set), a recommender system outputs predictions $\hat{y}_{ut}, t \in \mathcal{T}_{test,u}$. We first sort the predictions of the model in decreasing order and pick the top $N$ items (we use 20 in the experiments). We denote the set of candidate items by $\mathcal{T}_{20}$ and the item of rank $n$ by $t_n$. Additionally, suppose that the ground truth labels are $y_{ut} \in \{0, 1\}, t \in \mathcal{T}_{test,u}$, and denote $N_{test,u} = |\{y_{ut} = 1, t \in \mathcal{T}_{test,u}\}|$ as the number of ground truth items of the user $u$. We compute the $Recall$ metric as follows:

$$Recall(N) = \frac{|\{y_{ut} = 1, t \in \mathcal{T}_{20}\}|}{N_{test,u}} \tag{12}$$

$|\cdot|$ denotes the number of items in a set. $NDCG$ is short for Normalized Discounted Cumulative Gain. It is denoted as the ratio between Discounted Cumulative Gain ($DCG$) and ideal Discounted Cumulative Gain ($iDCG$), which are denoted as

$$DCG(N) = \sum_{n=1}^{N} \frac{\mathbb{1}_{\{y_{ut_n} = 1\}}}{log_2\{n + 1\}}$$

577    and

$$iDCG(N) = \sum_{n=1}^{N_{test,u}} \frac{1}{log_2\{n+1\}}$$

578    where $\mathbb{1}$ is the indicator function. $DCG$ takes the rank of the predictions and places more weight on
579    highly ranked items. While $iDCG$ is the ideal $DCG$ where all ground truth items $N_{test,u}$ are ranked
580    before the other items.

### D.1   Ablation and Hyper-Parameter Analysis

582    In this subsection, we perform the ablation and hyperparameter analysis. First, we study the effect of
583    different embedding aggregation functions. In our system, the final representation is the weighted
584    average of all embeddings (as defined in Eq. (2)). We consider two more intuitive choices for
585    embedding aggregation. In the first method, we only use the last latent embedding, and the final
586    representation is the average between the embedding and the highest order of latent embedding.
587    We denote this variant as FedGRec-last. The second choice is to concatenate all embeddings/latent
588    embeddings instead of summing them together. We denote this baseline as FedGRec-concat. We
589    test the three embedding methods on the Gowalla dataset and the results are summarized in Table 3
590    and Table 4. As shown in the table, FedGRec-last performs worse than FedGRec, especially in the
591    case of three latent embeddings. This shows that higher-order latent embeddings contain less useful
592    information compared to the lower ones. Regarding FedGRec-concat, we observe that it overfits
593    the training data when we set local iterations $\tau = 10$, so the results in Tables 3 and 4 choose $\tau = 1$.
594    Note that the latent embeddings during local training are fixed. As a result, latent embeddings work
595    as a constant bias term in the loss objective, and this makes the model overfit to the current latent
596    embeddings easier.

Table 3: NDCG@20 Comparison of Different Aggregation Methods on the Gowalla Dataset

| #Latent Embeddings | 1 | 2 | 3 |
|---|---|---|---|
| FedGRec | **0.1376** | **0.1412** | **0.1362** |
| FedGRec-last | **0.1376** | 0.1332 | 0.1246 |
| FedGRec-concat | 0.1266 | 0.1267 | 0.1254 |

Table 4: Recall@20 Comparison of Different Aggregation Methods over the Gowalla Dataset

| #Latent Embeddings | 1 | 2 | 3 |
|---|---|---|---|
| FedGRec | **0.1712** | **0.1695** | **0.1654** |
| FedGRec-last | **0.1712** | 0.1605 | 0.1493 |
| FedGRec-concat | 0.1502 | 0.1515 | 0.1494 |

597    Next, we investigate the effects of two hyperparameters: the number of users sampled per training
598    epoch $S$ and the number of local iterations $\tau$. The results are shown in Figure 3. First, as shown in the
599    top row of the figure, $S = 400$ gets the best performance, sampling more users per epoch accelerates
600    the early training stage, but it then slows down and converges to a sub-optimal point due to overfitting.
601    Next, as shown in the bottom row of the figure, the algorithm converges much faster when we set $\tau$
602    as 10 or 20 compared to when set as 1 or 2. This shows that our FedGRec benefits from performing
603    multiple local iterations. In other words, it is not necessary to update latent embeddings at each step,
604    and staled latent embeddings still help training.

