# OpenReview forum: "FedGRec: Federated Graph Recommender System with Lazy Update of Latent Embeddings"
_NeurIPS.cc/2022/Workshop/Federated_Learning — FL-NeurIPS 2022 Poster_

### Official Review · Reviewer_F7mC · 2022-10-15
**Extending GNN-based recommender systems to federated learning setting**

FedGNN ([48] in the paper) has adapted graph neural network (GNN) based recommender systems to the federated learning setting, but it only takes into account first-order user-item interactions.
This paper proposes to improve the performance of the recommender by take into account higher-order interactions, which are captured by latent embeddings in centralized GNN-based recommender system.
The main idea is to approximate the classic joint learning of users' and items' embedding in a centralized setting with two phases where partial updates occur in parallel at the clients and at the servers.
Each client stores its own user's embeddings, while the server stores all items' embeddings.
At each training round, the user retrieves a subset of items' embeddings and use them to first improve its own embeddings (considering items' embeddings fixed) and then items' embeddings (considering user's embeddings fixed). The updates to items' embeddings from each client are then aggregated at the server.
The approach shows good performance beating sota FedGNN and providing accuracy comparable with centralized approach. There is no theoretical guarantee of convergence.

I am not knowledgeable in modern recommender systems and I just trusted the authors in their description of the role of latent embeddings.
I find the proposed solution a quite natural way to extend the centralized approach to the FL setting and I do not see particularly creative choices (but I may be missing something given my limited expertise on recommender systems). Despite this, the idea is well executed and the results are promising.

There are two points I would have liked to be discussed/investigated more. The first one concerns privacy. The paper claims that high privacy is guaranteed by having each user maintaining its embeddings and by using secure aggregation techniques. But finally each user always queries for the embeddings of the items it interacted with, then the server could at least retrieve such information. Moreover, putting for a moment aside the use of secure aggregation technique, are we sure that clients' update do not leak some information about users' embeddings? The second one is related to the amount of information exchanged at each round. In the experiments each client retrieves the embeddings for all items it has interacted with plus embeddings for an additional 2k items. Given that the datasets count between 40k and 90k items, each client retrieves information between a significant percentage of the items. Moreover, 400 clients are selected at each round. This implies that at each communication round the number of client-item pairs considered is comparable (800k) with the actual number of interactions in the datasets (between 1 and 3 millions). I would have liked to see some experiments evaluating performance sensitivity to the information exchanged per round (e.g., can this be simply compensated by training for a longer time?).

Minor
- users keep its data -> their
- in the introduction, the term bias appears, but we do not know exactly what is denoted by bias.
- figure 1's caption: the sever randomly sample -> the server randomly samples
- p.3: resemble the design in the non-distributed setting, we let -> resembling? Inspired by? Similarly to what done?

---

### Official Review · Reviewer_ewkX · 2022-10-18
**Solid idea and empirical evaluation, but has significant overlap with prior work**

In short, I'm a bit torn on this paper. Considering this paper exclusively, without reference to prior literature, I think it's a good paper. The idea is simple but effective, and outperforms a number of other federated recommender system methods. Moreover, the idea is simple enough that pairing it with other important concepts in the literature (eg. Secure Aggregation) is effectively an entirely straightforward procedure.

I will caveat my primary reservation about this paper with the fact that I am not an expert in recommender systems. That being said, I think that the core idea in this work is the simple observation that in a federated recommender system, the server can store current item embeddings, and clients can store their local user embedding, and effectively "reconstruct" good user embeddings based on whatever item embeddings are sent to them by the server (see Figure 1).

This idea is exactly the core idea of "Federated Reconstruction: Partially Local Federated Learning" [Singhal et al., 2021]. There are obvious differences between these works, but they primarily reside in what tasks the various authors use to test this idea. While Singhal et al. do not provide empirical evaluations of their model in specifically graph-based recommender system tasks, the principle is the same (see Figure 1 of [Singhal et al., 2021]). Both papers note that they are SecAgg compatible, in part because their overall frameworks are roughly the same.

As a result, I have opted to give this a weak reject, with the caveat that I really do like the empirical work in this paper. There a wide variety of algorithms and tasks shown, with some useful ablation analysis.

---

### Decision · Program_Chairs · 2022-10-20

Accept (Poster)